# Zinc and Its Impact on the Function of the Testicle and Epididymis

**DOI:** 10.3390/ijms25168991

**Published:** 2024-08-19

**Authors:** Sergio Marín de Jesús, Rosa María Vigueras-Villaseñor, Edith Cortés-Barberena, Joel Hernández-Rodriguez, Sergio Montes, Isabel Arrieta-Cruz, Sonia Guadalupe Pérez-Aguirre, Herlinda Bonilla-Jaime, Ofelia Limón-Morales, Marcela Arteaga-Silva

**Affiliations:** 1Doctorado en Ciencias Biológicas y de la Salud, Universidad Autónoma Metropolitana, Ciudad de Mexico 09340, Mexico; uamarinsergio@gmail.com (S.M.d.J.); soniagaguirre@gmail.com (S.G.P.-A.); 2Laboratorio de Biología de la Reproducción, Instituto Nacional de Pediatría, Coyoacán, Ciudad de Mexico 04530, Mexico; rmvigueras@yahoo.com.mx; 3Departamento de Ciencias de la Salud, División de Ciencias Biológicas y de la Salud, Universidad Autónoma Metropolitana-Iztapalapa, Avenida San Rafael Atlixco 186, Ciudad de Mexico 09340, Mexico; cobe@xanum.uam.mx; 4Cuerpo Académico de Investigación en Quiropráctica, Universidad Estatal del Valle de Ecatepec, Av. Central s/n Valle de Anáhuac, Ecatepec de Morelos 55210, Mexico; joelhr19@hotmail.com; 5Unidad Académica Multidisciplinaria Reynosa-Aztlán, Universidad Autónoma de Tamaulipas, Calle 16 y Lago de Chapala, Aztlán, Reynosa 88740, Mexico; montesergio@gmail.com; 6Departamento de Investigación Básica, Instituto Nacional de Geriatría, Secretaria de Salud, Ciudad de Mexico 10200, Mexico; iarrieta@inger.gob.mx; 7Departamento de Biología de la Reproducción, Universidad Autónoma Metropolitana-Iztapalapa, Avenida San Rafael Atlixco 186, Ciudad de Mexico 09340, Mexico; lindabon35@gmail.com (H.B.-J.); ofelia.limon@yahoo.com (O.L.-M.); 8Laboratorio de Neuroendocrinología Reproductiva, Universidad Autónoma Metropolitana-Iztapalapa, Av. San Rafael Atlixco 186, Ciudad de Mexico 09340, Mexico

**Keywords:** Zn, testicle, spermatogenesis, epididymis, sperm maturation, redox system

## Abstract

Zinc (Zn) is an essential trace element; it exhibits a plethora of physiological properties and biochemical functions. It plays a pivotal role in regulating the cell cycle, apoptosis, and DNA organization, as well as in protein, lipid, and carbohydrate metabolism. Among other important processes, Zn plays an essential role in reproductive health. The ZIP and ZnT proteins are responsible for the mobilization of Zn within the cell. Zn is an inert antioxidant through its interaction with a variety of proteins and enzymes to regulate the redox system, including metallothioneins (MTs), metalloenzymes, and gene regulatory proteins. The role of Zn in the reproductive system is of great importance; processes, such as spermatogenesis and sperm maturation that occur in the testicle and epididymis, respectively, depend on this element for their development and function. Zn modulates the synthesis of androgens, such as testosterone, for these reproductive processes, so Zn deficiency is related to alterations in sperm parameters that lead to male infertility.

## 1. Introduction

Organisms require a variety of micronutrients, including vitamins, minerals, and trace elements, for survival. Among these, certain metals play significant roles in catalytic and protein structural processes [1]. Copper (Cu) and iron (Fe) participate in redox reactions through electron transfer or act as substrates for carrying out biochemical reactions. It is well established that a disruption in the regulation of these metals can result in oxidative stress [2].

Currently, there is considerable interest in investigating the potential of trace elements, which are considered to be elements that are required in amounts between 1 and 100 mg/d or less than 0.01% of the total body weight [3]. Some of these elements are essential metals for humans, including Fe, Cu, magnesium (Mg), manganese (Mn), selenium (Se), and Zn. Research has been conducted on the role of trace elements in disease prevention and their use as therapeutic agents. These are among the topics of interest for trace elements [4].

The significance of Zn in health and disease has predominantly been investigated from the standpoint of severe deficiency, manifesting with clearly discernible clinical indications. It is important to note that, in contrast to Fe or Cu, excessive Zn accumulation can result in disorders that are less severe than those associated with Fe or Cu excess [2].

A trace element is classified as essential based on three criteria: (1) presence in tissues, (2) total absence in the organism, which causes serious and irreversible damage, and (3) reduction in physiological function, which can be restored with adequate supplementation of that element [5]. Among these elements, Zn is of particular interest due to its extensive biochemical functions and physiological properties. Consequently, research into this element has been a significant area of focus in recent decades [6].

In contrast, there are non-essential metals, designated as toxic heavy metals, including arsenic (As) and cadmium (Cd), which lack any beneficial biological function. These metals can interact with proteins and DNA in a manner that is both strong and potentially damaging. This interaction can result in the excessive production of reactive oxygen species (ROS), which can cause specific damage to proteins or DNA. This damage is manifested as oxidation and/or conformational changes in proteins or DNA that lead to the development of serious diseases [2]. As a consequence of significant anthropogenic influences on the environment, humans may be exposed to these toxic metals through the inhalation of contaminated air, dietary intake of contaminated food, and drinking water [2]. The exposure to these toxic substances represents a global issue, as highlighted by numerous studies. Consequently, it is imperative to identify strategies for the prevention or mitigation of the damage caused by these environmental toxins which are detrimental to human health. There is a need to investigate the role of essential trace elements in the body, which have a significant impact on disease prevention. The study of essential trace elements allows for their importance in physiology, nutrition, clinical diagnosis, and human medicine to be highlighted.

This review presents a summary of the significant advances made in recent years regarding the essential trace element Zn. It covers the mechanisms that facilitate an understanding of its role in the reproductive system on the basis of biology, cellular mobilization through its primary transporters, and its main function as an inert antioxidant. Moreover, it details its function in maintaining redox balance and providing cellular protection. In the context of testicular development and spermatogenesis, this review highlights the importance of germ cells and their differentiation into spermatozoa, as well as the role of Zn in regulating androgens, such as testosterone, which facilitate this process (Table 1). Furthermore, the text describes the function of Zn in the epididymis for sperm maturation and its relationship with male fertility. This comprehensive approach enables the identification of the essential role of Zn in reproduction and its potential applications in medicine and male infertility.

## 2. Zn as a Trace Element

After Fe, the second most abundant trace element in the human body is Zn [13,14]. It belongs to the divalent metals in group 12 of the periodic table [2] and is one of the most important due to the enormous biological functions it performs. In addition, it is considered a micronutrient required for metabolism and sexual health [10,13]. It is possible that the medicinal use of Zn salts may have been known since ancient Roman times, when many drugs were known to contain Zn as an active ingredient [15]. However, its biological importance in human medicine was not fully understood until the 19th century and a century later [16]. In the 1930s, Zn was reported to be an important element for rodents. Thirty years later, its essentiality in humans was reported [17,18]. Zn deficiency may cause damage to various tissues; most of the research on its clinical impact and role in pathology is recent. In 1963, Zn deficiency was first described as a clinical phenomenon in humans after patients in Iran exhibited a syndrome characterized by Fe and Zn deficiency, known as anemia, along with hepatosplenomegaly, dwarfism, and hypogonadism. Subsequent studies on the role of Zn in biology had a significant impact, particularly regarding its involvement in proteins, enzymes, and molecular functions [17,18].

The studies that first recognized Zn as an essential nutrient began with the research by Jules Raulin on *Aspergillus niger*, which is an organism dependent on Zn. This was later corroborated by Prasad, who discovered Zn deficiency in humans between 1869 and 1963. From 1939 to 2006, the discoveries of Zn as a protein cofactor and the number of Zn-containing proteins were of great relevance to proteomics. Finally, the discoveries regarding Zn ions in cellular regulation, signaling, and molecular functions lack a clearly defined beginning and remain open to future implications [18].

The discovery of a nutritional deficiency in Zn in humans was met with controversy, given that it was related to pathological states. In 1974, the National Research Council of the National Academy of Sciences declared Zn as an essential element for human health. In 1978, the United States Food and Drug Administration (FDA) established that Zn must be included in nutrition [19]. In adult humans, Zn concentrations in the body range from 1.4 to 2.3 g, with significant distribution in all tissues [15]. Specifically, 49.5% of Zn is found in muscle, 36.7% in bone, 7.6% in skin, and 6.2% in all other tissues [20]. The U.S. National Library of Medicine Toxicology Data Network (TOXNET) database indicates that the oral lethal dose of 50% Zn is considered to be 3 g/kg of body weight. Although it is considered an element with relatively low toxicity, if intake exceeds 150 mg/d over prolonged periods, as it can cause severe damage [2,21].

Biochemical studies of Zn have continued, and now, the physiological functions dependent on this element have been extensively reviewed [18]. These include its role in wound healing phases, such as in the functionality of immune cells to regulate the immune response [22,23]. Consequently, a deficiency in this element results in adverse effects on health, including alterations to the immune system, metabolism, and other vital functions [23,24].

Zn is regarded as a crucial regulator among inert redox metal ions, including sodium (Na), potassium (K), and Mg. Furthermore, Zn is capable of functioning as an intracellular messenger, a property it shares with calcium (Ca). Consequently, the regulatory pathway of Zn interacts with Ca²⁺ ion signaling [5]. Zn plays a pivotal role in numerous physiological processes, including cell cycle regulation, apoptosis, the organization of DNA, RNA, proteins, lipids, and carbohydrates, as well as the stability of cell membranes [11,25]. In prokaryotic cells, approximately 83% of Zn-containing proteins perform the function of enzymatic catalysis. In eukaryotic cells, 47% of Zn is involved in biological functions, such as catalytic reactions, 44% in DNA transcription, 5% in protein transport, and 3% in signaling pathways, in addition to providing stability to the plasma membrane structure [2,26].

## 3. Zn Transport

In mammals, Zn absorption occurs in the small intestine. Most cells obtain Zn from the interstitial fluid, which is derived from the blood plasma. In this fluid, Zn mainly binds to proteins, such as serum albumin and α2-macroglobulin. Subsequently, it is distributed to tissues. In humans, the concentration of total Zn in plasma/serum ranges between ~12 and 17 μM [27,28].

Due to its hydrophilic nature, Zn is unable to cross the plasma membrane or the membranes of intracellular compartments [2]. Consequently, the mobilization of Zn at the cellular level is directed by two primary Zn transporter families: One is the Zn importing family (ZIP), which facilitates the transport of Zn into the cytosol that has been identified to comprise 14 distinct ZIP types. The other family of Zn transporters is the Zn transporter (ZnT). These proteins facilitate the export of Zn from the cytosol, with 10 types of ZnT proteins having been identified [28,29,30]. Upon entering the cell through these transporters, Zn is directed to the endoplasmic reticulum (ER), mitochondria, Golgi apparatus, and proteins [16]. At the cellular level, approximately 40% of Zn is found in the nucleus and 50% in the cytoplasm, while the remainder is associated with membranes and proteins [16,24,31]. Concerning cytosolic Zn, it binds to apoproteins thioneins to form MTs [16,32]. Together with the transporters ZIP and ZnT, MTs regulate the supply of excess Zn and control its distribution at deficiency sites [16,30] (Figure 1).

Other membrane proteins, including some types of voltage-dependent Ca channels, transient receptor channels, acetylcholine receptors, and glutamate receptors, are also involved in Zn mobilization [32]. A currently unidentified transporter may buffer Zn within a silencer, which subsequently translocates the metal into intracellular storage sites, such as the ER, Golgi apparatus, mitochondria, or lysosomes. However, the mechanism by which intracellular and extracellular Zn is stored is unknown [33].

## 4. Zn as an Inert Antioxidant

Within the organism, a multitude of biological processes dependent on redox reactions, which consist of the transfer of electrons from one chemical substrate to another, and the uncontrolled “leak” of electrons from these reactions, leads to the formation of ROS [34,35]. It is crucial to maintain redox balance as the inhibition of or alteration in antioxidant enzymes may lead to an overproduction of ROS, which, in turn, causes oxidative stress and cellular damage, ultimately leading to the development of chronic diseases [7,34,36,37].

To maintain redox balance, antioxidants can either directly or indirectly catalyze ROS and thereby protect cells. There are several endogenous enzymatic antioxidants, including superoxide dismutase (SOD), catalase (CAT), glutathione peroxidase (GPx), and thioredoxins (TRx), as well as non-enzymatic antioxidants such as vitamins (retinol and α-tocopherol), minerals (Se and Zn), flavonoids, phenols, and others. These antioxidants are effective in counteracting free radicals and neutralizing oxidants, thereby preventing cellular damage [37,38].

The role of Zn in relation to oxidative damage has been extensively studied [7], as it is an element with inert antioxidant properties. Unlike other bioactive metals, such as Fe or Cu, Zn does not undergo redox reactions. It is redox-inactive as it cannot donate or receive electrons and it is always present in its oxidative state Zn (+2) in biological systems. Consequently, it is not considered an antioxidant in the strict sense [2,8,39]. However, Zn ligands are capable of interacting with a large number of proteins and enzymes to fulfill a wide variety of important functions, including modulation in the redox system [8].

Proteins that contain Zn in their structural sites are designated as metalloproteins. The human genome has been sequenced, and proteomic approaches have been employed independently to identify more than 1600 proteins that require Zn to carry out their biological functions [40]. Metalloproteins are classified into three major groups: (1) metalloenzymes, (2) MTs, and (3) gene regulatory proteins [16].

A total of 300+ enzymes have been identified that require Zn for their function [11], representing the only metal found in all six classes of enzymes, including oxidoreductases, hydrolases, transferases, isomerases, lyases, and ligases [40]. Zn plays a role in three crucial functions in enzymes: (1) catalytic function, where it directly participates in enzymatic catalysis; (2) coactive function, where it enhances or diminishes catalytic function in conjunction with catalytic Zn; and (3) structural function, where it provides stability to the quaternary form of oligomeric enzymes [31]. The binding of Zn to proteins is dependent on the specific amino acid composition of the protein. Four amino acids, cysteines (Cys), histidines (His), aspartic acid (Asp), and glutamic acid (Glu), have been identified as the primary binding partners for Zn. The Cys-Zn interaction occurs at the structural Zn sites, while the interaction with Glu and Asp acids occurs at the catalytic Zn sites. Finally, the His-Zn interaction occurs at both Zn binding sites [28,32,41].

In the antioxidant system, Zn plays a unique role in enzymes [7]. The metalloenzyme SOD is a primary homodimeric antioxidant enzyme that catalyzes the disproportionation of superoxide anion (O_2−_) to hydrogen peroxide (H_2_O_2_) and molecular oxygen (O_2_), thereby reducing the toxicity of ROS by converting a highly reactive radical to a less harmful one [7,42,43]. In mammals, three distinct isoforms of SOD have been identified: the cytosolic Cu/Zn-SOD dimer (SOD1), the mitochondrial matrix Mn-SOD (SOD2), and the extracellular tetrameric EC-SOD (SOD3) [44,45]. Each homodimer of SOD1 contains an active site that binds to a catalytic Cu ion and a structural Zn ion. These metal-binding, disulfide bond formation, and dimerization processes represent post-translational modifications that are essential for the acquisition of the mature form of SOD1, which is characterized by proper folding and enzymatic activity [43,46,47]. Although SOD1 is primarily located in the cytoplasm, it can also be found in other cellular compartments, including the nucleus, ER, and mitochondria [47].

Structurally, SOD1 is a 153-amino acid polypeptide folded into an eight-stranded β-barrel motif. This site is formed by two major loops: one electrostatic and one Zn loop that forms the active site. Zn binding occurs within a coordination complex comprising three His residues and one Asp, linked by a disulfide bridge within the unit. One of the histidine residues that binds to Zn also binds to copper. Therefore, copper is responsible for catalytic activity, while Zn plays the roles of the regulator of catalytic activity and the stabilizer of the metalloenzyme structure. The binding of Cu in SOD1 is carried out by chaperones, although the precise mechanism of Zn binding in these metalloenzymes remains unknown [47,48,49].

One of the crucial mechanisms of Zn is its ability to strongly bind to MTs. Approximately 20% of cytosolic Zn binds to apoprotein thioneins to form MTs [2,16]. MTs are ubiquitous 67 kDa proteins that are rich in Cys and form complexes with transition metals such as Zn, Cu, Fe, and Cd, among others. As a result, they play an important role in balancing excessive amounts of metal ions and providing cellular protection. Consequently, they represent one of the main detoxification mechanisms for heavy metals [50]. It has been demonstrated that up to seven Zn ions can bind to MT molecules, with this binding exhibiting a higher stoichiometry than with other Zn proteins. This is evidenced by the fact that four Zn ions bind to eleven Cys residues in the α domain and three bind to nine Cys residues in the β domain [51]. The coordination of thiolate Zn in these domains is necessary for physiological processes. The β domain is more labile than the α domain, with ROS clearance and essential metal homeostasis being carried out by the β domain, whereas the α domain has a function in heavy metal detoxification [28,51,52]. MTs are known to correlate with the redox potential of reduced glutathione/glutathione disulfide (GSH/GSSG) [53]. MTs are primarily located in the cytoplasm and can be transported to the nucleus, where they engage in various functions, including DNA protection from oxidative damage and interactions with transcription factors [28].

In contrast, gene regulatory proteins are designated as nucleoproteins and fulfill their direct function through DNA replication and transcription. The most important of these DNA-binding proteins are “Zn fingers” [54]. The first discovery of the structure of Zn fingers was in the 1980s, and subsequent decades have increased the knowledge of the number of proteins with Zn fingers, indicating and reinforcing their role in the structure of transcription factors in the human genome [55].

The thiolate Zn groups are sensitive to the cellular redox state; therefore, under oxidative conditions, the release of Zn ions from MTs generates Zn signals to carry out the expression of enzymes necessary for the elimination of ROS [28,56]. The rise in cytosolic Zn is identified by several transcription factors, which then modulate antioxidant defenses or even promote cellular functions such as proliferation and apoptosis [8,28].

Zn is a potent inducer of MTs [57] and is recognized by metal regulatory transcription factor (MTF-1), a protein containing six Zn-fingered domains. Zn binds to MTF-1, and the complex then translocates to the nucleus and binds to the metal response elements (MREs) in the promoter region, thereby increasing the transcription of thioneins and some Zn transporters, such as ZnT-1 [7,43,58,59].

Nuclear factor erythroid 2 (Nrf2) is also a Zn-sensitive transcription factor and is a protein Zn finger. Upon Zn binding, Nrf2 translocates to the nucleus and binds to antioxidant response elements (ARE), thereby activating the upregulation and transcription of various antioxidant proteins [7,60]. These include glutamate–cysteine ligase, which modulates the rate of GSH synthesis. GSH is an intracellular antioxidant that predominates in many organisms, protecting cells from free radicals, lipid hydroperoxides, toxic xenobiotics, and heavy metals [61,62]. Zn has been demonstrated to exert a dual effect in neutralizing free radicals. This occurs directly via the action of GSH or indirectly with the involvement of a cofactor of GPx [7]. Consequently, MTF-1 and the Nrf2 factor depend on Zn to modulate the expression of MTs, antioxidants, and other proteins, which directly neutralize free radicals. Furthermore, Zn plays an important role with its protective capacity by presenting anti-apoptotic properties [7,59,63] (Figure 2).

Oxidative stress is one of the factors leading to cell apoptosis. Zn has been demonstrated to possess anti-apoptotic properties; however, the mechanisms by which Zn protects against apoptosis are still not clearly understood [64,65]. Research indicates that Zn activates the cellular protection process through the modulation of apoptosis-regulating proteins, including caspase-3, the Bcl-2 family of proteins, and the BAX gene [66,67]. Caspase-3 is a pivotal mediator of programmed cell death, orchestrating the initiation, execution, and termination of apoptosis. Bcl-2 proteins form a complex network that promotes cell survival, while other proteins, such as BAX, induce apoptosis [67].

## 5. The Relevance of Zn in the Testes during Spermatogenesis

A review of the literature on the functions of Zn in reproduction reveals that it plays an essential role in male reproductive potential, particularly in its participation during spermatogenesis, the maturation of spermatozoa during their journey in the epididymis, and furthermore, in events preceding fertilization within the female reproductive tract [12].

Spermatogenesis, the process of sperm production, occurs within the seminiferous tubules of the testicle. The presence of Zn is of great importance for this event [9]. ZnT and ZIP proteins are responsible for the supply and balance of this element, as the survival of germ cells and the replacement of protamines during the spermatogenic process depend on Zn [68]. It has been demonstrated that mitochondria can provide Zn via transporters and that Zn exerts a regulatory effect on mitochondrial function in germ cells [12].

ZIP proteins are located on the plasma membrane of Sertoli cells, where they acquire Zn from circulation. Subsequently, ZnT is responsible for exporting Zn to developing germ cells. The expressions of ZIP and ZnT are specific to Leydig cells, Sertoli cells, and various germ cells [69]. Zn is present in the nuclei of germ cells and plays a role in spermatogenesis, including processes such as self-renewal, differentiation, and proliferation. Zn accumulation increases during the condensation and meiosis periods in spermatocytes, and Zn also facilitates DNA packaging in spermatids [69,70].

The replacement of histones by protamines represents a significant protective mechanism that is employed during the spermatogenic process to safeguard against the deleterious effects of DNA damage [71]. During the final stages of differentiation in spermatids, Zn is incorporated into the nucleus, where it binds with protamines to form disulfide bonds and stabilize the chromatin structure [72,73,74]. Thymidine kinase is an important enzyme for DNA synthesis. While it is not a Zn metalloenzyme, Zn regulates the transcription of this enzyme by binding Zn-dependent proteins to the promoter region of the gene [75]. Consequently, Zn deficiency results in alterations to the thymidine kinase, ultimately leading to failure at the early stages of spermatogenesis. This can manifest as impaired spermatogonial proliferation and arrest, as well as the death of germ cells that fail to reach the morphologically complete spermatozoa stage [12]. Zn finger proteins also play a significant role in spermatogenesis. These transcription factors are essential for germ cell proliferation and differentiation [12,76]. In spermatozoa, Zn binds to the sulfhydryl groups of Cys of the outer dense fibers, which are protein fibers that envelop the axoneme of the flagellum for its motility. This binding of Zn is necessary to protect the flagellum from premature oxidation [72,77].

Leydig cells are the primary site of testosterone production and development in the testicle, occurring in two distinct growth phases: the fetal and pubertal/adult populations. Zn plays a pivotal role in Leydig cells by participating in the production, storage, and transport of testosterone, the primary hormone regulating spermatogenesis [78]. The ZnT-8 transporter is localized to the mitochondria of human and mouse Leydig cells. This protein facilitates the transport of Zn from the cytoplasm into the mitochondria. The transport of Zn mediated by ZnT-8 regulates the phosphorylation of acute steroidogenic protein (StAR) for testosterone production [79]. Zn finger proteins also play a pivotal role during spermatogenesis, with a primary focus on germ cell proliferation and differentiation. Furthermore, the expression of Zn finger proteins in Leydig cells is necessary for testosterone synthesis [76]. The expression of MTs, SOD, and GPx in the testis is of great importance in counteracting ROS [80,81,82]. Therefore, Zn plays a crucial role in regulating these enzymes and preventing oxidative damage in Sertoli cells, Leydig cells, and germ cells. Consequently, various transporters are involved in the accumulation of Zn during the spermatogenic process. It has been demonstrated that a Zn deficiency results in the downregulation of several proteins, including ZIP, ZnT, Zn finger proteins, and antioxidant enzymes. This, in turn, leads to the downregulation of steroidogenic proteins and testosterone biosynthesis in Leydig cells, alterations in seminiferous tubules, and the failure or arrest of spermatogenesis [83]. Zn deficiency in the testicle has been demonstrated to induce oxidative stress, stimulate pro-apoptotic signaling (Bax and caspase-3), and suppress anti-apoptotic signals in germ cells [84].

## 6. The Relevance of Zn in the Epididymis and Sperm Maturation

Once spermatozoa have left the testicle and entered the epididymis, they have already developed and differentiated morphologically. However, they are still considered functionally immature, as at this point, they cannot yet recognize, bind, or penetrate oocytes [85]. The epididymis is an essential male sex organ as it is vital for reproduction and provides protection to maturing sperm [86,87]. Furthermore, it is an organ rich in Zn, especially in the epithelial region of the cauda in the apical part of the principal cells and the epididymal lumen. Therefore, it has been considered that the presence of this element is involved in the maturation of spermatozoa [88]. The epididymis absorbs the higher concentration of Zn bound to the dense fibers during spermatogenesis. Following the removal of Zn, sulfhydryl groups (Zn-thiol) are oxidized, hardening the outer dense fibers. Therefore, eliminating Zn is necessary to provide progressive motility to the spermatozoa [77].

In the epididymis, Zn plays a pivotal role, as the majority of proteins secreted by this organ depend on Zn to provide functionality to spermatozoa [89,90]. The transfer of proteins secreted by the epididymis to spermatozoa is mediated by epididymosomes, which are Zn-dependent and function as enhancers for the interaction between epididymosomes and spermatozoa [91,92]. Furthermore, the epididymis is an androgen-dependent organ, particularly with regard to the conversion of testosterone to 5α-dihydrotestosterone (5α-DHT). This conversion is carried out by the enzyme 5α-reductase, which is Zn-dependent for this process [93].

During the maturation process of sperm in the epididymis, spermatozoa are susceptible to oxidative damage because they are unable to synthesize antioxidant enzymes. Consequently, the epididymis is responsible for protecting against oxidizing agents through a large battery of enzymes [86]. GPx 5 and TRx are among the most abundant antioxidant enzymes in the epididymis [85,86,94]. Their expression is regulated by Zn in addition to acting as cofactors of SOD and strongly inducing the increase in MTs for the scavenging of free radicals [43]. Moreover, during epididymal transit, spermatozoa are vulnerable to free radicals and are targets for lipoperoxidation due to the high concentration of polyunsaturated fatty acids in their plasma membranes. This susceptibility may influence the ability to acquire maturation, including motility and chromatin condensation. Zn integrates with membranes by interacting with membrane-associated proteins, including myelin-associated glycoproteins and lipoproteins, via sulfhydryl groups [95]. This interaction stabilizes the spermatozoa membrane, providing a protective layer and thus increasing fertilization potential. Additionally, Zn reduces lipoperoxidation in spermatozoa [96,97].

## 7. Zn and Its Role in Sperm Quality

Infertility is defined as the biological inability of a couple to produce a pregnancy during at least one year of unprotected sexual intercourse. Approximately 15% of couples experience infertility, with approximately 30–50% of these cases being related to male factor infertility. This encompasses a range of conditions, including structural abnormalities (e.g., varicocele), endocrine disorders, genital tract infections, immune deficiencies, obesity, aging, alcohol consumption, and tobacco, as well as exposure to environmental factors such as pesticides, and radiation [82]. It is clear that a deficiency in trace elements can have a negative impact on reproductive function and sperm quality [9]. Human semen contains various trace elements, including Zn, Cu, Ca, Mg, Mn, and Se, which are essential for reproductive processes [72,76]. Therefore, deficiencies in these essential elements can result in alterations in sperm quality and male fertility [9].

In the seminal fluid, Zn plays a pivotal role in maintaining spermatozoa viability. Consequently, a deficiency or decrease in Zn concentration in seminal plasma will be reflected in a low sperm count, as well as low quality and morphological alterations in spermatozoa [79,97]. The absence of or deficiency in Zn in the testicle leads to hypogonadism and alterations in the development of secondary sexual characteristics [19,79,98]. Furthermore, it is associated with oxidative damage to lipids, alterations in the transcription and translation of DNA, and testicular cell protein damage [99].

A deficiency in Zn uptake by spermatogonia impairs differentiation and reduces the number of germ cells that become spermatozoa, which is associated with a reduction in the sperm concentration. In addition, there is evidence of severe damage to the histoarchitecture of the seminiferous tubules and a reduced testicular weight [100]. Zn deficiency impairs the proliferation and differentiation of Leydig cells, as well as their function in testosterone synthesis and secretion. This, in turn, leads to impaired spermatogenesis and poor sperm quality [79,101]. One potential cause of a low sperm concentration is the increase in ROS during spermatogenesis, which can lead to germ cell apoptosis. Consequently, Zn exerts a modulatory effect on the upregulation of antioxidant enzymes and the regulation of germ cell apoptosis [102].

## 8. Zn in Other Organs of the Reproductive System

The prostate is one of the sexual organs known to contain high concentrations of Zn, with levels reaching 150 µg/g, which is three times higher than that in any other soft tissue. Furthermore, prostatic fluid contains 500 µg of Zn/mL [97,103,104]. Consequently, the prostate releases large concentrations of Zn into the seminal plasma during ejaculation, which enhances sperm motility [105]. Another significant role of Zn in the prostate is to provide antimicrobial properties, thereby protecting spermatozoa during ejaculation [106].

In the seminal vesicles, Zn plays a pivotal role in coagulation and semen stability, as well as in improving sperm motility for successful fertilization [107]. Semen is viscous in consistency. However, if Zn concentrations are low, this causes high secretion by the seminal vesicle, resulting in a hyperviscous consistency. This is associated with a decrease in sperm motility, morphological abnormalities, and a low seminal volume [108].

The presence of Zn in seminal plasma is of great significance in the context of seminal stress, as oxidative stress can lead to apoptosis, and Zn can prevent this process [109]. Studies conducted in humans indicate that the concentration of Zn in semen (0.23–2.30 mg/mL) is related to normospermia samples, which demonstrates that this element plays a vital role in sperm morphology [12]. Consequently, Zn supplementation has been demonstrated to enhance semen quality in males with suboptimal Zn levels, resulting in augmented seminal volume and sperm count, as well as improved sperm motility and morphology [110].

## 9. Conclusions and Future Research Directions

In recent times, studies have been conducted to ascertain the pivotal role of Zn in the human body and to identify the potential consequences of Zn deficiency or absence on human health. Due to its antibacterial properties, it has a regulatory function as a catalytic and structural component of numerous proteins and enzymes. In addition to its anti-apoptotic properties, it has been demonstrated that it meets the characteristics of an essential trace element for biological organisms. In male reproductive health, Zn is a crucial element for reproductive success, starting from the formation of spermatozoa from the first germ cells and testosterone synthesis to spermatogenesis. Its importance in sperm maturation during their journey through the epididymis, in sperm survival after ejaculation, during the journey in the female reproductive tract, and even in fertilization is why Zn deficiency leads to a low fertility rate. Zn is a particularly intriguing factor, as it is regarded as a protective element that can compete with other metals that do not fulfill vital functions. Its role in counteracting oxidative stress through different mechanisms provides fundamentals for future implications of therapeutic use in male reproduction, as well as supplementation and treatment in various diseases.

In the future, it will be interesting to know whether Zn may provide new additional benefits in physiological or pathological conditions in male reproduction. These encompass cellular signaling, hormonal regulation, and the protection and interaction with proteins and enzymes that regulate these processes. Conversely, several researchers have demonstrated that a deficiency in this element can result in male infertility. In accordance with these findings, the medical community recommends Zn administration or supplementation within the appropriate context as a dietary supplement.

Updates to the mechanisms of Zn provide valuable tools for future research, not only in the context of the reproductive system, but also at the neurological and pathological levels since its properties comprise an important basis, and there is still more to understand about its biology and chemical interactions that could be correlated with major discoveries for human health.

## Figures and Tables

**Figure 1 ijms-25-08991-f001:**
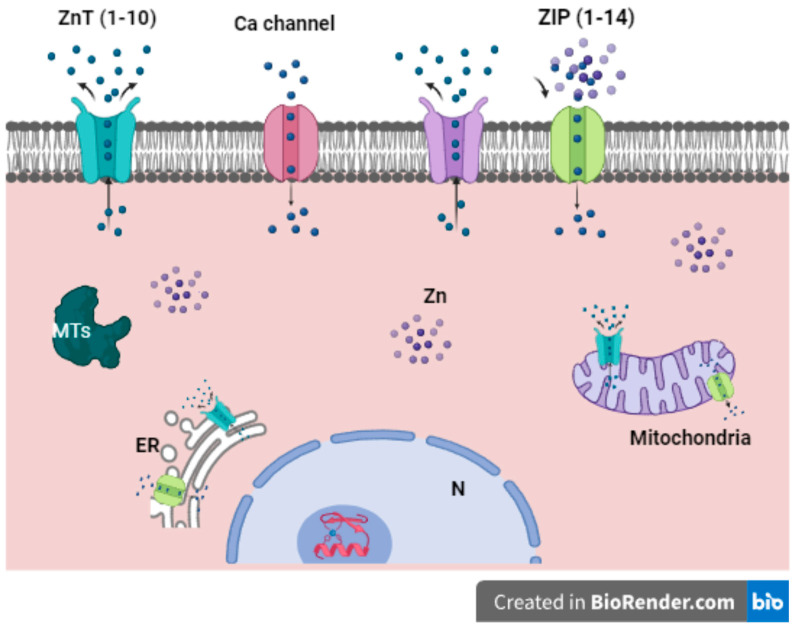
A schematic representation of Zn transporters is presented below. The mobilization of Zn towards the cytosol is mediated by ZIP, while cytosolic Zn is transported to the outside of the cell by ZnT. The cell plasma membrane and cellular organelles present different isoforms of ZnT and ZIP, which, together with MTs and Ca channels, regulate the cellular Zn concentration. Zn is targeted and stored in cellular compartments, such as the ER and mitochondria, for specific functions. MTs: metallothionein; N: nucleus (created in BioRender.com).

**Figure 2 ijms-25-08991-f002:**
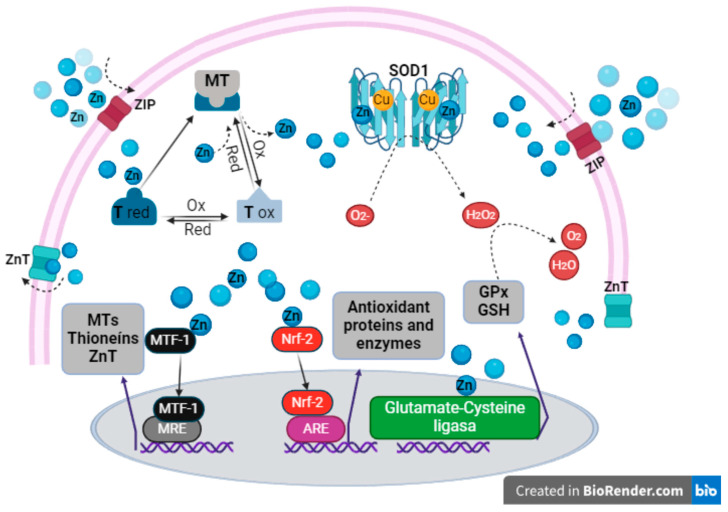
The antioxidant mechanisms of Zn are as follows: Zn enters the cell through ZIP and binds to thioneins to form metallothionein (MT); free Zn is recognized by the MTF-1 factors, translocates to the nucleus, binds to MREs, and activates the expression of thioneins and ZnT. The Nrf2 factor also recognizes free Zn, translocates to the nucleus, and binds to ARE to activate the expression of antioxidant enzymes. Zn binds to the apoprotein SOD, thereby regulating its antioxidant function. Ultimately, Zn is transported by ZnT to the outside of the cell. Antioxidant response elements (AREs) are also DNA sequences that bind to Nrf-2. GSH is a tripeptide that acts as an antioxidant. GPx is an enzyme that catalyzes the reduction of hydrogen peroxide (H_2_O_2_) to water (H_2_O). O_2_- is the chemical formula for a superoxide anion. ZIP: Zn import proteins, ZnT: Zn transport proteins, MT: metallothionein, T red: reduced thionein, T ox: oxidized thioneine, SOD1: superoxide dismutase 1, MTF-1: metal regulatory transcription factor. MREs are DNA sequences that bind to the nuclear factor erythroid 2 (Nrf-2) transcription factor. The figures were created using BioRender.

**Table 1 ijms-25-08991-t001:** A comparison of the contributions about the participation of Zn in reproduction.

The Contributions of Previous Reviews	The Contribution of the Present Review
The biochemical aspects and mechanisms of Zn against oxidative stress [7].	The properties and biology of Zn and its mechanisms of action.
Zn is an inert antioxidant, and it participates in oxidative stress [8].	The interaction of Zn with proteins and enzymes, such as MTs, metalloenzymes, and gene regulatory proteins.
The deficiency in Zn in the testicle and seminal plasma and its role in male infertility [9].	The importance of Zn in male fertility, from hormonal regulation by androgens and testicular and spermatogenic functions, as well as its importance in germ cells for differentiation into spermatozoa and its role in the epididymis.
The role of Zn in the hyperactivated motility of human spermatozoa [10].	Zn participates in sperm survival and functionality, and it is necessary for successful fertilization.
The contribution of Zn to quality sperm parameters [11].	Zn is a promoter of improvement in the quality of sperm parameters in male fertility.
The role of Zn in the testicle, prostate, and seminal vesicle and in sperm viability [12].	The role of Zn and its regulation in the testicle and the epididymis for the optimization of sperm production and maturation.

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
