# Peer review of "Zinc and Its Impact on the Function of the Testicle and Epididymis"

_ijms, 2024, doi:10.3390/ijms25168991_

Round 1

Reviewer 1 Report

Comments and Suggestions for Authors

The scientific content of the ms (review) is very interesting, and I thus favor its acceptance in the Int. J. of Mol. Sci. I am sure that the article will attract the intense interest of scientists working in the area of the elucidation of molecular mechanisms of the action of trace elements in biology, and especially those concerning Zn(II); bioinorganic chemists will also find this review interesting. Also, I do believe that the article will receive a respectable number of citations in the near and far future. Zn(II) is a very important trace element for living organisms because it exhibits a variety of physiological properties and biochemical functions.

The review has been organized in sections. Section 1 is an introduction clarifying the concepts of trace and essential elements. Section 2 refers to Zn(II) as a trace element. Zn(II) plays a central role in cell cycle regulation, apoptosis, organization of vital biomolecules (DNA, RNA, proteins, lipids, carbohydrates, etc.). Section 3 describes the transport processes of the metal ion. The mobilization of Zn(II) at the cellular level and within the body is achieved mainly by two transporter families, the Zn importing family (ZIP) and the Zn transporters (ZnT). Section 4 deals with the role of Zn(II) in relation to oxidative damage; it is not considered as an antioxidant in the strict definition of the term. However, the biological ligands at Zn(II) can interact with many proteins and enzymes and sometimes modulate redox systems. Section 5 reviews the function of Zn(II) in reproduction. It is an essential metal ion in male reproductive potential, particularly in its participation during spermatogenesis and maturation of spermatozoa. Section 6 explains the pivotal role of Zn(II) in maintaining male fertility. Deficiency of Zn(II) concentration in seminal plasma leads to a low sperm count, as well as to low quality and morphological changes in spermatozoa. Section 7 outlines the function of Zn(II) in prostate. Finally, Section 8 concludes the role of Zn(II) in the human body and explains the consequences of Zn(II) deficiency in human health. The quality of the three figures is high and the reference list covers the topic under study satisfactorily.

Based on the above mentioned, I am more than happy to recommend acceptance of this fine article in IJMS. Since the present work is a review I do not have scientific points to raise. Minor points/comments/suggestions are listed below:

(1) The English of the ms should be improved. There are several grammatical and syntax errors, e.g., "membranas" instead of "membranes".

(2) According to IUPAC rules, Zn is a group 12 (and not II-B element).

(3) The same number (5) has been assigned to two different sections; this should be corrected.

(4) According to my opinion, a new section would be added briefly describing the role of Zn(II) in Alzheimer’s disease.

Comments on the Quality of English Language

The English of the ms should be improved.

Author Response

We thank Reviewer 1 for their accurate observations to our work. We are sure that these insights improve the quality of our manuscript.   

Question 1: The English of the ms should be improved. There are several grammatical and syntax errors, e.g., "membranas" instead of "membranes".

Answer: We fully agree with the reviewer’s comments, and all sections of the article have been addressed. As suggested, we reviewed the writing and grammar to correct all the reviewer’s observations. The manuscript was sent for English editing, and the English writing and grammar were checked throughout all sections of the manuscript by a native English speaker.

Question 2: According to IUPAC rules, Zn is a group 12 (and not II-B element).

Answer: The update was reviewed according to IUPAC rules, and we agree that Zn belongs to group 12 of the periodic table. This was checked in the manuscript and the correct information was added.

Question 3: The same number (5) has been assigned to two different sections; this should be corrected.

Answer: We appreciate your comments. We reviewed all sections of the manuscript and corrected the assignment of number 5 in both sections.

Question 4: According to my opinion, a new section would be added briefly describing the role of Zn (II) in Alzheimer’s disease.

Answer: We appreciate your comment and opinion on the role of Zn (II) in Alzheimer's disease. However, our focus in this review is on reproduction. Nevertheless, we consider it important to conduct research on the neurological role of Zn (II) in the future.

Reviewer 2 Report

Comments and Suggestions for Authors

The manuscript is of general interest. The following comments should help further improve the quality of the work:

1-The manuscript should be improved in terms of the usage of English. Authors are advised to get their manuscript edited by a native English speaker or by a Professional English Editing Service.

2-Abstract should be improved by including the major findings of the review, where applicable quantitatively too.

3-Provide a graphical abstract.

4-Using too long paragraphs should be avoided.

5-The Introduction is too brief and does not present the state-of-the-art, and should be expanded.

6-The originality of the present work should be effectively established. Please include a table in the Introduction section tabulating the latest reviews in this domain. In that table, please include the various features covered by these reviews and the present work. This should efficiently help highlight the contribution of the present review against the existing literature.

7-The Article structure lacks a clear section on methodology. Perhaps to improve the article, the authors should use the PRISMA methodology to analyze the literature.

8-The criticality of the work should be enhanced by including critical discussions.

9-Please make sure all the units will be presented in compliance with the SI System. For instance, please use "d" for "day", etc. 

10-Please change "8. Conclusion” to “Conclusions and future research directions". Accordingly, please elaborate on the future research needs in this domain.

11-The practical implication of the present review should be discussed as well.

12-It is advisable to add DOIs for some references

Comments on the Quality of English Language

The manuscript should be improved in terms of the usage of English. Authors are advised to get their manuscript edited by a native English speaker or by a Professional English Editing Service.

Author Response

We thank Reviewer 2 for their accurate observations to our work. We are sure that these insights improve the quality of our manuscript.   

The manuscript is of general interest. The following comments should help further improve the quality of the work:

Question 1: The manuscript should be improved in terms of the usage of English. Authors are advised to get their manuscript edited by a native English speaker or by a Professional English Editing Service.

 Answer: We completely agree with the reviewer's comments, and all sections of the article have been addressed. As suggested, we reviewed the writing and grammar to correct all the reviewer's observations.

Question 2: Abstract should be improved by including the major findings of the review, where applicable quantitatively too.

 Answer: In accordance with the reviewer's suggestions, we revised the abstract to include the main findings, enhancing the content's fluency in the review.

Question 3: Provide a graphical abstract.

 Answer: In the new version of the manuscript a comprehensive graphical abstract is provided.

 Question 4: Using too long paragraphs should be avoided.

 Answer: We completely agree with the reviewer's comments, and all sections of the article have been revised to avoid long paragraphs.

Question 5: The Introduction is too brief and does not present the state-of-the-art, and should be expanded.

Answer: We agree with the reviewer's comments, and we have revised the introduction section to expand on the information as recommended.

 Question 6: The originality of the present work should be effectively established. Please include a table in the Introduction section tabulating the latest reviews in this domain. In that table, please include the various features covered by these reviews and the present work. This should efficiently help highlight the contribution of the present review against the existing literature.

 Answer: We are grateful for the observation and suggestion. We concur with this recommendation and will include the proposed table to highlight the existing literature in relation to the present work. 

Question 7: The Article structure lacks a clear section on methodology. Perhaps to improve the article, the authors should use the PRISMA methodology to analyze the literature.

Answer: We appreciate the suggestion and comment regarding this point. However, we consider that this work is not a systematic review. Our work is focused mainly in the cellular and molecular mechanisms of zinc on male reproductive system. Nonetheless, it would be important in the future to undertake a more comprehensive and systematic review of the topic since an interventional or therapeutic vision, as there are still areas to be developed and more mechanisms to be explored to help us address and understand new and emerging research.

 Question 8: The criticality of the work should be enhanced by including critical discussions.

 Answer: Thank you so much for this suggestion. We have included a new section about future research directions to discuss the needs in the field and integrate the zinc effects with others biological systems.

Question 9: Please make sure all the units will be presented in compliance with the SI System. For instance, please use "d" for "day", etc. 

 Answer

We completely agree with the reviewer's comments, and all sections of the article have been revised as suggested. The writing and grammar were reviewed to address all the reviewer's observations in accordance with the International System of Units (SI).

Question 10: Please change "8. Conclusion” to “Conclusions and future research directions". Accordingly, please elaborate on the future research needs in this domain.

Answer:

We appreciate the observation and have incorporated the reviewer's suggestion. We agree that it is important to highlight future research directions related to the topic discussed in the manuscript.

Question 11: The practical implication of the present review should be discussed as well.

 Answer: Yes, we agree with this suggestion. In the new section about future research directions, we have included the biological implication of the administration of zinc supplementation within the appropriate context as a dietary supplement.

Question 12: It is advisable to add DOIs for some references

 Answer: We agree with the reviewer regarding the use of DOIs and have added them to the references. We appreciate their interest and observations.

Round 2

Reviewer 2 Report

Comments and Suggestions for Authors

Can be accepted in its present form.